# Oocyte Quality, In Vitro Fertilization and Embryo Development of Alpaca Oocytes Collected by Ultrasound-Guided Follicular Aspiration or from Slaughterhouse Ovaries

**DOI:** 10.3390/ani12091102

**Published:** 2022-04-25

**Authors:** Leandra Landeo, Michele Zuñiga, Teddy Gastelu, Marino Artica, Jaime Ruiz, Mauricio Silva, Marcelo H. Ratto

**Affiliations:** 1Laboratory of Reproductive Biotechnologies, Faculty of Engineer Sciences, Universidad Nacional de Huancavelica, Huancavelica 09001, Peru; llandeoj@unam.edu.pe (L.L.); sthfany_zl@hotmail.com (M.Z.); roni_rockperu@hotmail.com (T.G.); marino.artica@unh.edu.pe (M.A.); jaime.ruiz@unh.edu.pe (J.R.); 2Vicepresidencia de Investigación, Universidad Nacional de Moquegua, Moquegua 18001, Peru; 3Department of Veterinary Science and Public Health, Faculty of Natural Resources, Universidad Católica de Temuco, Temuco 4780000, Chile; masilva@uct.cl; 4Núcleo de Investigación en Producción Alimentaria, Universidad Católica de Temuco, Temuco 4780000, Chile; 5Department of Animal Science, Faculty of Veterinary Sciences, Universidad Austral de Chile, Valdivia 5090000, Chile

**Keywords:** alpaca, oocytes, ovum pick-up, in vitro embryo production, blastocyst

## Abstract

**Simple Summary:**

We compared the morphological quality and the in vitro developmental competence of cumulus–oocyte complexes (COCs) from live vs. slaughtered alpacas. COCs were recovered using: (i) manual aspiration in slaughterhouse ovaries (*n* = 15 females), or (ii) ovum pick-up (OPU) in live superstimulated alpacas (*n* = 13). COCs recovered were morphologically evaluated. Grade I–III COCs were in vitro matured and fertilized afterwards using fresh alpaca spermatozoa. Zygotes were in vitro cultured for seven days. The proportion of COCs recovered was similar between groups, but the mean number of COCs collected from individual ovaries was greater (*p* < 0.05) in slaughterhouse ovaries. A higher (*p* < 0.05) percentage of grades III and IV and a lower (*p* < 0.05) percentage of grade I COCs was obtained using OPU. Although, no significant difference in the percentage of cleavage and morula formation was observed between groups, the number of blastocysts, regarding cleavage and COCs collected, was higher (*p* < 0.007 and *p* < 0.0002, respectively) in the OPU group. However, the total number of blastocysts per female did not differ between groups. The recovery rate and morphological quality of COCs was significantly higher when follicles were manually aspirated from slaughterhouse alpaca ovaries; however, a greater developmental potential was observed in oocytes obtained from live alpaca.

**Abstract:**

The morphological quality and the in vitro developmental competence of cumulus-oocyte complexes (COCs) collected from in vivo or slaughtered alpacas was compared. COCs were recovered from ovarian follicles using: (i) manual aspiration in ovaries of alpacas (*n* = 15) sacrificed at a local slaughterhouse, or (ii) transrectal ultrasound-guided follicular aspiration (or ovum-pick-up, OPU) in live alpacas (*n* = 13) 4 days after the administration of an ovarian superstimulation protocol (200 UI eCG). COCs recovered from both groups were morphologically evaluated and graded. Grade I to III COCs were in vitro matured for 26 h and in vitro fertilized afterwards for 20 h using fresh alpaca epididymal spermatozoa. Presumptive zygotes from both groups were in vitro cultured for 7 days. The proportion of COCs recovered over the total number of follicles punctured was similar between groups, but the mean number of COCs collected from individual ovaries was greater (*p* < 0.05) in slaughterhouse ovaries. A significantly higher (*p* < 0.05) percentage of low-quality COCs (grades III and IV) and a lower (*p* < 0.05) percentage of grade I COCs was obtained using OPU. The number of blastocysts, regarding cleavage and COCs collected, was higher (*p* < 0.007 and *p* < 0.0002 respectively) for COCs collected by OPU; however, the total number of blastocysts per female did not differ between groups. We can conclude that the recovery rate and morphological quality of COCs was significantly higher when follicles were manually aspirated from slaughterhouse alpaca ovaries; however, a statistically higher developmental potential was observed in oocytes collected by OPU from live alpaca donors.

## 1. Introduction

The development of in vitro embryo production (IVP) in South American camelids (SACs) is in a relatively earlier stage compared to other domestic mammals, such as cattle or sheep, where more research has been developed during the last decades [1,2]. This technology has been widely adopted by those animal industries and it is used routinely as a potential strategy for genetic improvement. However, in camelids, many unsolved deficiencies of in vitro maturation, fertilization and embryo culture procedures need yet to be addressed and overcome to standardize a consistent and efficient IVP process [3,4].

Several South American countries (Peru, Chile, Bolivia and Argentina) have large populations of SACs, and other countries, such as USA, New Zealand, Australia, Germany, who have imported a large number of llamas and alpacas from South America in the last three decades, also have animals with high genetic merit. There is no doubt that the application of assisted reproductive techniques, such as in vivo and in vitro embryo production, using oocytes from live donors will be a valuable tool to increase the number and quality of these herds. Although studies of in vivo embryo production have been reported in alpacas [5] and llamas [6,7], there is still a gap in the knowledge of the in vitro production of embryos in these species [3], and it is necessary to develop and establish consistent procedures to collect oocytes from live animals and to successfully use them in an in vitro embryo production program.

Pioneer studies of in vitro maturation and fertilization in South American camelids used ovaries and testes collected from slaughterhouses as the source of gametes [8,9]. The information generated from those early studies allowed standardizing of in vitro maturation and fertilization protocols in alpacas and llamas, resulting in the birth of the first llama offspring from an IVP embryo 20 years later [10]. 

Along with the main steps of IVP of embryos, other techniques, such as transvaginal ultrasound-guided follicular aspiration, need to be improved to allow consistent recovery rates of high-quality oocytes from valuable females. The first studies to describe the use of transvaginal follicular aspiration or ovum pick-up (OPU) in SACs were conducted in llamas with the purpose of collecting oocytes for ultrastructure studies [11] or to synchronize the follicular wave emergence for natural insemination [12]. Then, two additional studies in llamas reported in detail the effect of transvaginal follicular aspiration on oocyte recovery rate, in vitro [9] and in vivo [13] maturation potential and embryo production. However, to date there is just one study [14] describing the use of transvaginal follicular aspiration in alpacas, but no data of embryo development was reported.

The use of oocytes collected from slaughterhouses was a valuable source of biological material to set up all the procedures involved in IVP of embryos in these species as it was done in other commercial species, such as bovine or sheep; however, camelids abattoirs are very limited in South America and non-existent in North America or Europe. Moreover, it has been demonstrated that COCs collected from slaughterhouse ovaries have very heterogeneous developmental potential due to undergoing atresia in a proportion of the follicular population, because of the physiological status of the females. Therefore, their development potential has resulted in low blastocyst rates [8,15,16,17]. On the contrary, COCs obtained by OPU from in vivo llamas, during the growing or static phase of follicle development [18] or after oocyte in vivo maturation [13] have resulted in a blastocyst rates greater than 30 and 20%, respectively.

Few studies of alpaca embryo IVP [17,19] have reported the production of morulas or blastocysts after in vitro fertilization, however, oocytes from those studies have been collected from slaughterhouse ovaries [17] or from live donors using a surgical approach [19]. OPU has been described in several studies in llamas [9,11,13,20] but only one of them [13] reported high morula and blastocyst rates after IVF of in vivo matured COCs collected from llamas superstimulated using FSH or eCG.

Although, South American camelid embryos have been produced from slaughterhouse ovaries [8,17,21] and in vivo animals [13,18,19,22] there is not a direct comparison of the developmental competence between oocytes collected from slaughterhouse ovaries versus in vivo animals using OPU.

Therefore, the purpose of the present study was to evaluate the efficiency of alpaca in vitro embryo production, comparing the developmental potential of COCs recovered in vivo by ovum pick-up versus those collected from slaughterhouse ovaries. 

## 2. Materials and Methods

All chemicals and reagents were purchased from Sigma Chemical Co. (St. Louis, Mo, USA) unless stated otherwise. All procedures related to embryo IVP were conducted at the Laboratory of Reproductive Biotechnologies, Escuela Profesional de Zootecnia, National University of Huancavelica, Huancavelica, Peru (12°47′ South and 74°58′ West, 3680 m.a.s.l.). The study was conducted in accordance with the Declaration of Helsinki, and approved by the Institutional Ethics Committee (Universidad Austral de Chile Bioethical committee, 395/2020).

### 2.1. Collection of Ovaries from Slaughterhouse 

Thirty ovaries were collected from mature non-pregnant alpacas (*n* = 15) during the breeding season (November–April) from the local abattoir (Huancavelica, Peru) and transported to the laboratory in 37 °C phosphate buffered saline (PBS, Gibco, Invitrogen Corporation, Grand Island, NY, USA) supplemented with penicillin (100 IU/mL) and streptomycin (100 µg/mL) within 1 h after slaughter. Follicles from 3 to 6 mm in diameter were aspirated using a 21-gauge needle attached to a sterile 10 mL syringe containing PBS (37 °C) supplemented with penicillin (100 IU/mL), streptomycin (100 μg/mL) and 0.3% bovine serum albumin. Follicular aspirates were transferred to 10 mL conical tubes and allowed to settle for 10 min. Then, the cellular precipitate was aspirated with a sterile plastic pipette and transferred to a 100 mm plastic Petri dish where the oocytes were located using a stereoscope (15× magnification).

### 2.2. Cumulus Oocyte Complexes Collection Using Ovum Pick-Up (OPU) 

The alpaca herd was kept at the Alpaca Research Station of the University of Huancavelica. Mature non-pregnant alpacas (*n* = 15) ≥4 years of age and weighing 65–70 kg, were used as oocyte donors during November–April (breeding season). Females were maintained in separate pens with access to natural pasture and supplemented with hay and water ad libitum.

Females were subjected to a synchronization protocol to control and stimulate ovarian follicular growth. In brief, ovaries were scanned daily using transrectal ultrasonography (Aloka SSD-500 scanner) coupled with a 7.5 MHz linear array transducer. Females with the presence of a follicle ≥7 mm in diameter were given an intramuscular administration of 50 μg of GnRH analogue (gonadorelin acetate, day 0: GnRH treatment) to induce ovulation. Afterwards, females were scanned every 12 h to determine ovulation. Ovulated females were treated with 200 IU of eCG. On day 4, females were subjected to caudal epidural anesthesia (2.5 mL of lidocaine; Bimeda-MTC Animal Health Inc., Cambridge, ON, Canada) and 5 minutes later to transvaginal ultrasound-guided follicle aspiration using a 5.0 MHz convex-array ultrasound transducer coupled to a 19-gauge needle. Follicles ≥5 mm were aspirated using a regulated vacuum pump at a flow rate of 22 mL/min into a 50 mL conical tube containing phosphate buffered saline (PBS) at 37 °C, supplemented with 0.3% bovine serum albumin (BSA) and 0.1% EDTA. Aspirates were transferred to Petri dishes for COC collection using a stereomicroscope. 

### 2.3. Morphological Classification of Cumulus Oocyte Complexes (COCs)

The COCs collected from slaughterhouse ovaries and transvaginal aspiration were identified, evaluated and categorized using a stereomicroscope (magnification of 15×) coupled with a heater plate at 37 °C. A classification scale ranging from grade I to IV according to the number of cumulus cell layers and the appearance of the oocyte cytoplasm was used as previously described [9]. In brief, COCs were classified as grade I (≥4 layers of compacted granulosa cells tightly surrounding the oocyte and homogeneous ooplasm), grade II (partial layers of granulosa cells scattered around the oocyte and homogeneous ooplasm), grade III (denuded, mostly absence of granulosa cells but homogeneous ooplasm) and grade IV (expanded/degenerated, pyknotic granulosa cells and vacuolated ooplasm).

### 2.4. In Vitro Maturation and Fertilization of Cumulus Oocyte Complexes

Only grades I, II and III COCs were used for in vitro maturation. COCs collected from slaughterhouse ovaries and transvaginal aspiration were separately washed in TCM-199 (Gibco, USA) and then in vitro matured in groups of 10–15 in 50 μL drops of TCM-199 supplemented with 10% heat-treated fetal calf serum (FCS, Gibco), 0.2 mM sodium pyruvate, 0.5 μg/mL FSH, 1 μg/mL estradiol-17β and 25 µg/mL gentamycin. Microdrops were covered with mineral oil and cultured for 26 h at 38.5 °C in a humidified atmosphere of 5% CO_2_ in air. 

After in vitro maturation, all oocytes were fertilized in vitro using epididymal sperm obtained by the sperm washing technique described by [17]. In brief, 7 alpaca epididymis were collected from mature males at the local slaughterhouse and transported on ice to the laboratory within 30 min of sacrifice. Epididymis tails were dissected and placed in Sperm-TALP (2 mM CaCl_2_, 3.1 mM KCL, 0.4 mM MgCl_2_.6H_2_O, 0.3 mM NaH_2_PO_4_.H_2_O, 21.6 mM lactic acid (sodium salt; 60% w/w syrup), 100 mM NaCl, 1 mM Na pyruvate, 25 mM NaHCO_3_, 10 mM HEPES, 0.6% (w/v) BSA and Fraction V). Sperm were recovered under stereomicroscopy by puncturing and squeezing the tissue, and aspirating the secretion obtained with a 1 mL syringe attached to a 30-gauge needle. Sperm were then pooled and evaluated for motility. Samples (1.5 mL) with ≥75% progressive motility were suspended in Sperm-TALP and centrifuged twice at 300 × *g* for 6 minutes. After the first centrifugation the sperm pellet was resuspended in Sperm-TALP and after the second centrifugation the final pellet was resuspended with Fert-TALP (2 mM CaCl_2_, 3.2 mM KCL, 0.5 mM MgCl_2_.6H_2_O, 0.4 NaH_2_PO_4_.H_2_O, 11 mM lactic acid (sodium salt; 60% w/w syrup), 114 mM NaCl, 0.2 mM Na pyruvate, 25 mM NaHCO_3_, 20 mM Penicillamine 10 mM hypotaurine, 1 mM epinephrine, 0.6% (w/v) fatty acid-free BSA, 10 μg/mL heparin and 50 mg/mL gentamicin sulfate) to a final concentration of 1.5 × 10^6^ spermatozoa/mL (80% sperm motility). Groups of 10 matured COCs were washed with PBS supplemented with BSA, and then transferred into a multi-well dish. Each well contained 50 μL of spermatozoa suspension. Gametes were co-incubated at 38.5 °C in air with 5% CO_2_ and high humidity for 18–20 h. 

### 2.5. In Vitro Culture of Embryos

After in vitro fertilization, presumptive zygotes were denuded by gentle pipetting for 3 min, washed 3 times in SOFm-Hepes and finally cultured in groups of 20–25 in a multi-well dish with 250 µL of culture medium. Each well contained SOFm-IVC medium supplemented with 10% fetal serum, 0.13% BME essential amino acids, 0.063% MEN non-essential amino acids, 1 mM L-glutamine, 10% insulin and 50 µg/mL gentamicin sulfate. Zygotes were cultured at 38.5 °C in a humidified atmosphere of 5% CO_2_ in air for 8 days. Culture medium was changed every 24 h using SOFm-SFB supplemented with 2 mM glucose. Cleavage, morula and blastocyst rate were evaluated at 48, 96 and 120 h after in vitro fertilization.

### 2.6. Statistical Analysis

Number of COCs recovered and in vitro matured, percentage of recovered COCs and their morphological quality, cleavage, morula and blastocyst rate were compared between both groups using the Wilcoxon–Mann–Whitney and Kruskal–Wallis test. Data were analyzed using the Statistical Analysis Systems package (SAS, v8.02). Probabilities ≤0.05 were considered significant.

## 3. Results

Two females from the OPU group were eliminated from the analysis because of limitations in the procedure of follicular aspiration; therefore, 13 females (26 ovaries) were aspirated using this technique.

The proportion of COCs recovered over the total number of follicles punctured were similar between groups, but the mean number of COCs collected from individual ovaries was greater (*p* < 0.05) in the slaughterhouse ovaries than that of the OPU group (Table 1). The morphological quality of CCOs recovered from slaughterhouse ovaries was higher than that of COCs recovered by OPU. A significantly higher (*p* < 0.05) percentage of low quality CCOs (grades III and IV), and a lower (*p* < 0.05) percentage of grade I COCs was obtained using OPU (Table 1, Figure 1). 

Although, there was not a significant difference in the percentage of cleavage and morula formation between groups, the number of blastocysts regarding cleavage and COCs collected was higher (*p* < 0.007 and *p* < 0.0002, respectively) for COCs obtained by OPU (Table 2); however, the total number of blastocysts per female did not differ between groups (Table 2, Figure 1).

## 4. Discussion

To the best of our knowledge, this is the first report of in vitro blastocyst production using COCs collected by ovum pick-up from alpaca live donors. Based on the results of this study, although a comparatively higher number of oocytes were collected from slaughterhouse ovaries, a higher blastocyst rate (89%) was observed in the OPU group demonstrating the higher developmental potential of oocytes recovered from synchronized live donors compared to those obtained from slaughterhouse ovaries. The above, demonstrates the potential use of this technique to multiply genetically superior alpaca females, positively impacting the alpaca fiber industry and the economy of local communities.

In the case of llamas and alpacas, the collection of oocytes from live donors has been traditionally performed using a surgical approach. Ventral or lateral laparotomy procedures have been used to access and expose the ovaries to allow their manipulation and a direct manual aspiration of antral follicles [3]. Comparatively few studies have used ultrasound-guided transvaginal ovum pick-up; however, the advantages of this latter method upon the former have been clearly stated [3].

In the present study, collection technique significantly affected the number of COCs recovered. In the case of slaughterhouse ovaries, higher recovery rates are probably due to the ready access to manually punctured ovarian follicles equal or greater than 3 mm in diameter. The mean number of COCs collected from each slaughterhouse ovary (8.5 ± 0.7) was similar to the 7.6 COCs/ovary previously reported by [14], but higher than others [9,17]. COC recovery rate may be influenced by several factors, some dependent on the animal, such as age, body condition, physiological status, follicle number and size [23], and others dependent on the technique used (needle size, vacuum pressure, non-treated vs. hormonal treated females, single aspiration vs. follicle flushing [24]). 

COCs recovery rate by OPU was lower than manual aspiration of slaughterhouse ovaries, which could be due to live animals’ manipulation, response to the ovarian superstimulation protocol and technique difficulties. In this sense, [14] reported an average of 6 COCs/ovary, in alpacas treated with 700 IU of eCG, which is three times higher than the present study, probably due to a greater development of follicles available for transvaginal aspiration

In previous llama studies [9,13] the use of gonadotrophins for ovarian superstimulation increased the number of follicles and facilitated the recovery of mature oocytes by OPU technique. High and consistent COCs recovery rate shows that OPU is a useful and effective tool in several animal species [8,14,24,25] including llamas and alpacas [8,13,14,18].

Collection technique not only influenced the number of COCs recovered, but also significantly impacted their morphological quality. Thus, ultrasound-guided transvaginal follicle aspiration statistically reduced the quality of COCs recovered. The number of medium and low-quality grade COCs (III and IV) was significantly higher when the OPU technique was used in live alpacas, with roughly 10 and 20% of all COCs recovered considered as excellent or good quality, respectively. This deleterious effect of OPU technique has been previously reported in alpacas [14], llamas [11] and cattle [24]. In this regard, it has been reported that the amount of vacuum pressure used for the procedure is one of the key factors to be controlled in several animal species [24] to reduce the damage of COCs. However, the proportion of COCs recovered (CCOs recovered/follicles punctured) was similar between groups; and even more, a 75.8% of collection rate in the OPU group is similar to those described in previous llama studies [9,13] and higher than those results reported in cattle (60% on average [26,27]). In this regard, the location and attachment of COCs with the granulosa internal lining of the follicular wall should also be considered, since there are mayor differences between animal species in type of attachment of the COCs to wall granulosa (hillock or broad connection) and in the location of the COCs within the follicle in relation to the ovulation site. These aspects have been thoroughly described in horses, cattle and llamas, with important interspecies differences described [28]. 

The developmental potential of COCs collected by the two techniques was significantly different regarding cleavage and blastocyst formation. The cleavage rate was higher than that reported by [13] for in vivo matured oocytes collected from superstimulated llamas using FSH and eCG protocols. 

Although the development up to the morula stage was similar between groups, blastocyst rate was significantly greater for COCs collected by OPU. It has been reported that those COCs collected from slaughtered animals have a wide and heterogeneous range of quality and developmental potential, since they come from females with different reproductive statuses. This usually hampers the repeatability of the results in these types of studies. On the other hand, COCs collected from live donors through OPU technique are usually obtained from non-dominant follicles which have not reached a final stage of development or maturation. Thus, their developmental potential it is usually higher compared to that of oocytes obtained from slaughtered females.

In the present study, the developmental potential up to the blastocyst stage of COCs collected by OPU was higher than those reported in llamas recently by [13,18] for in vivo matured oocytes. The present blastocyst rate (89.2%) obtained with COCs collected by OPU is the highest reported up to date for any camelid. 

Total number of blastocysts produced by female donors was similar for both collection techniques (2.9 and 2.5 for slaughterhouse ovaries vs. OPU, respectively) even though the initial number of COCs collected was roughly a third for OPU vs. slaughterhouse ovaries. The above clearly shows the potential of using OPU associated with IVP in alpacas as a valuable biotechnology to accelerate the reproduction of highly valuable females. OPU has been used in llamas and alpacas as a non-invasive, safe, reliable and efficient tool to collect oocytes from live donors. The use of this technique in conjunction with IVP of embryos is essential to establish an accelerated program of genetic improvement. Moreover, oocytes collected by OPU can be also used to produce embryos by intracytoplasmic sperm injection (ICSI), bipartition, nuclear transfer and cloning [20,21]. 

Based on our results we can conclude that the recovery rate and morphological quality of COCs was significantly higher when follicles were manually aspirated from slaughterhouse alpaca ovaries. On the contrary, a statistically higher developmental potential was observed in oocytes collected by OPU from live alpaca donors.

## Figures and Tables

**Figure 1 animals-12-01102-f001:**
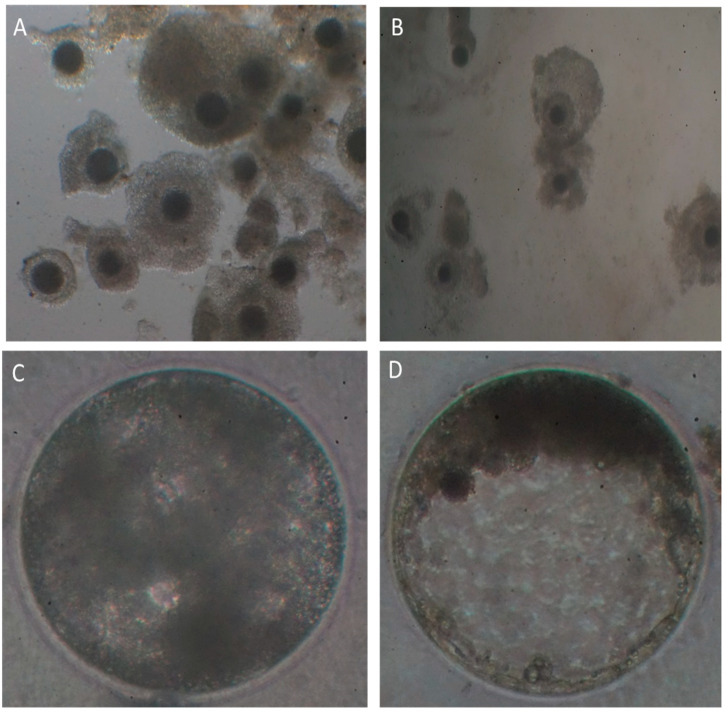
Representative images of COCs collected by OPU (**A**) from live alpacas or by manual follicular aspiration from slaughterhouse ovaries (**B**), early (**C**) and expanded blastocyst (**D**) produced in vitro.

**Table 1 animals-12-01102-t001:** Total number of punctured follicles, COCs recovered and quality of oocytes collected from slaughterhouse ovaries or in vivo donors using OPU in alpacas (%) (mean ± SD).

Groups	No. Ovaries	CCOs Recovered/Follicles Punctured	CCOs Recovered/Ovary	Quality of COCs (%)			
				I	II	III	IV
Slaughterhouse (*n* = 15)	30	128/154 (83.1%)	8.5 ± 0.7 ^a^	72/128 ^a^ (56.3)	19/128 (14.8)	21/128 ^a^ (16.4)	16/128 ^a^ (12.5)
OPU (*n* = 13)	26	50/66 (75.8%)	3.3 ± 2.9 ^b^	5/50 ^b^ (10)	11/50 (22)	21/50 ^b^ (42)	13/50 ^b^ (26)

^a,b^ = different superscripts indicate significant differences (*p* < 0.05).

**Table 2 animals-12-01102-t002:** Total number of COCs recovered and in vitro matured, cleavage, morula and blastocyst rate and total number of blastocysts produced by female using oocytes collected from slaughterhouse ovaries or in vivo donors using transvaginal follicular aspiration in alpacas.

Groups	Ovaries (*n*)	COCs Recovered	COCs Matured	Cleavage	Morula/ Cleavage	Blastocysts/ Cleavage	Blastocysts/ COCs	Blastocysts/ Female
Slaughterhouse (*n* = 15)	30	128 ^c^	112 ^c^	62/112 (55.4%)	55/62 (89%)	44/62 ^a^ (71%)	44/112 ^c^ (39.3%)	2.9
OPU (*n* = 13)	26	50 ^d^	37 ^d^	35/37 (94.6%)	33/35 (94%)	33/35 ^b^ (94%)	33/37 ^d^ (89.2%)	2.5

^a^ vs. ^b^ *p* = 0.007; ^c^ vs. ^d^ *p* = 0.0002.

## Data Availability

We choose to exclude this statement.

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
