# Peer review of "Oocyte Quality, In Vitro Fertilization and Embryo Development of Alpaca Oocytes Collected by Ultrasound-Guided Follicular Aspiration or from Slaughterhouse Ovaries"

_animals, 2022, doi:10.3390/ani12091102_

Round 1

Reviewer 1 Report

The paper "Oocyte quality, in vitro fertilization and embryo development of alpaca oocytes collected by ultrasound-guided follicular aspiration" supplies interesting information about IVP and OPU results in this species. However, after carefully examination of the manuscript, there are some considerations that need to be rectified. 

  1. First of all, the title suggests the article is about OPU only, when in stead a comparison between IVP and OPU/IVF is provided. I suggest to change the title slightly to clarify this.
  2. Line 115: “Thirty ovaries individually identified …”. I would suggest leaving out the “individually identified”
  3. Was an ethical committee required by national legislation? If so, please provide the ethical committee and the EC number provided by them.
  4. Was pain management provided before or after the OPU procedure? It is suggested to describe the management (other than the synchronisation protocol) of the alpaca mares a bit more elaborately.
  5. Line 174/175: It is described that semen samples were centrifuged twice, and that the final sperm pellet was resuspended. Firstly, in what exactly where the sperm samples centrifuged (e.g. SpermTALP, was it Single Layer Centrifugation, …?). Secondly, was the sperm pellet also resuspended after the first centrifugation, or only after the second centrifugation?
  6. On numerous occasions the abbreviation CCO is used in stead of COC. Also, the abbreviation OPU is often not used, in stead the authors write “transvaginal aspiration” more frequently. It is suggested to address this to be more consistent.

Author Response

Reviewer #1: The paper “Oocyte quality, in vitro fertilization and embryo development of alpaca oocytes collected by ultrasound-guided follicular aspiration” supplies interesting information about IVP of OPU results in this species. However, after carefully examination of the manuscript, there are some considerations that need to be rectified.

Response: We are glad our research generated a positive perception.

Reviewer #1:  First of all, the title suggests the article is about OPU only, when in stead a comparison between IVP and OPU/IVF is provided. I suggest to change the title slightly to clarify this.

Response: We have modified the title according to the previous suggestion.

Reviewer #1: Line 115: “Thirty ovaries individually identified…” I would suggest leaving out the “individually identified”

Response: We have eliminated “individually identified”.

Reviewer #1: Was an ethical committee required by national legislation? If so, please provide the ethical committee and the EC number provided by them.

Response: The information requested has been included in the M&M section.

Reviewer #1: Was pain management provided before or after OPU procedure? It is suggested to describe de management (other than the synchronization protocol) of the alpaca mares a bit more elaborately.

Response: The information requested has been included in the M&M section.

Reviewer #1: Line 174/175: It is described that semen samples were centrifuged twice, and that the final sperm pellet was resuspended. Firstly, in what exactly where the sperm samples centrifuged (e.g. SpermTALP, was it Single Layer Centrifugation, …?). Secondly, was the sperm pellet also resuspended after the first centrifugation,  or only after the second centrifugation?

Response: All the information requested has been included in the M&M section. Individual clarifications have been made for each question.

Reviewer #1: On numerous occasions the abbreviation CCO is used in stead of COC. Also, the abbreviation OPU is often not used, in stead the authors write “transvaginal aspiration” more frequently. It is suggested to address this to be more consistent.

Response: We agree with this comment. We have revised the entire manuscript and made all the corrections indicated.

Reviewer 2 Report

The article is interesting and shows results that may further help in future research.

The main question addressed by the research is if COCs obtained by OPU vs. collection from slaughterhouse are equally efficient to develop embryos in vitro.

The topic is relevant in the field because it describes another useful method to obtain COCs and develop embryos in vitro, which could be favorable to improve reproductive indexes in camelids, which are low.

The present work allows comparing the efficacy of the OPU method vs. COCs collection from slaughterhouse ovaries, to develop embryos in vitro. There is similar previous research but in one case COCs were obtained from laparotomy (Trasorras et al., 2012) and in another study from the same authors of the present study, which reports COCs collection by OPU in camelids (llamas), results are not compared to slaughterhouse collection and in vitro maturation was not performed (Berland et al., 2011).

The methodology is adequate and well described. A scheme of the methodology and showing photographs of the recovered COCs of the different groups and the developed blastocysts would improve the manuscript, and make reading easier, if possible.

Conclusions are consistent with the research and address the main question proposed

The references are appropriate.

Author Response

Reviewer #2:
The article is interesting and shows results that may further help in future research.
The main question addressed by the research is if COCs obtained by OPU vs. collection from slaughterhouse are equally efficient to develop embryos in vitro.
The topic is relevant in the field because it describes another useful method to obtain COCs and develop embryos in vitro, which could be favorable to improve reproductive indexes in camelids, which are low.
The present work allows comparing the efficacy of the OPU method vs. COCs collection from slaughterhouse ovaries, to develop embryos in vitro. There is similar previous research but in one case COCs were obtained from laparotomy (Trasorras et al., 2012) and in another study from the same authors of the present study, which reports COCs collection by OPU in camelids (llamas), results are not compared to slaughterhouse collection and in vitro maturation was not performed (Berland et al., 2011).

Response: We greatly appreciate the Reviewer’s comments about this manuscript.

Reviewer #2:
The methodology is adequate and well described. A scheme of the methodology and showing photographs of the recovered COCs of the different groups and the developed blastocyst would improve the manuscript, and make reading easier, if possible.

Response: Description of materials and methods have been revised and improved. Thus, we consider that the incorporation of a scheme it is not necessary.  Photographs of COCs obtained by OPU or manual follicular aspiration from slaughterhouse ovaries, and blastocysts produced in vitro have been included in figure 1 as suggested.

Reviewer #2:
Conclusions are consistent with the research and address the main question proposed.
The references are appropriate.

Response: We appreciate the Reviewer’s comments.